# Evaluating Doppel's impact on Anxiety and Focus amongst adults with ADHD

**Georgina Bartlett**[1]*, **Daniel Frings**[1], **Eddie Chaplin**[2]

1 School of Applied Sciences, London South Bank University, London, England, 2 Institute of Health and Social Care, London South Bank University, London, England

* bartleg2@lsbu.ac.uk

## Abstract

Attention Deficit Hyperactivity Disorder (ADHD) is the most commonly diagnosed psychiatric disorder in children. Amongst adults, it is often underdiagnosed and associated with comorbidities including anxiety. This study presents a trial evaluating the efficacy of *Doppel*, a wrist-worn wearable that provides vibrations linked to one's heart rate to improve symptoms of anxiety and poor focus amongst young adults with ADHD. Young adults (aged 18–25) used either an active or comparator Doppel for 8 weeks, completing measures of anxiety and focus at baseline, 4 weeks, and 8 weeks. Participants in both groups experienced a reduction in anxiety and an increase in focus across the trial duration. No superiority for vibrations linked to one's heart rate was found. Whilst the current study cannot determine a specific mechanism of action, the findings provide some promising initial evidence as to the potential for direct-to-consumer digital health products to be useful in symptom management amongst young adults with ADHD.

## Author summary

ADHD is often underdiagnosed in adults, and treatment options typically can include medication and psychological interventions aimed at self-management of symptoms. Digital health technologies such as wearables may provide a valuable tool for symptom management by helping to reduce anxiety. In this study we tested one such wearable, Doppel a wrist worn device that provides heartbeat like vibrations to the wrist. We compared an active Doppel vs a comparator to examine whether it could help reduce anxiety and increase focus in young adults (18–25) with ADHD. Participants used the Doppel device for 8 weeks and completed measures of anxiety and focus at regular intervals during the trial. We found that anxiety levels decreased, and focus increased in both groups, however there was no advantage of the active Doppel over the comparator. Adherence was high, with participants using the device on average seven hours a day. The findings suggest that wearable technology is well tolerated and may be an effective option for anxiety reduction for adults with ADHD.

**Data Availability Statement:** The data supporting the findings of the study is openly available on The Open Science Framework https://osf.io/mz5ps/.

**Funding:** The project is funded via the European Regional Development Fund (GB, DF, EC) grant

number: 23R15S00024, funding body the Greater London Authority: https://www.london.gov.uk/. Funds are administered by South Bank University Enterprise Ltd. Doppel contributed funding for participant incentives. The funders had no role in study design, data collection and analysis, decision to publish, or preparation of the manuscript.

**Competing interests:** The authors have declared that no competing interests exist.

Attention Deficit Hyperactivity Disorder (ADHD) is a psychiatric disorder associated with symptoms of hyperactivity, inattention and/ or impulsivity, that persist in two or more settings and cause at least moderate psychological/social/educational impairment [1]. The current paper presents the results of a pre-registered trial which tests the efficacy of a wearable tech solution (*Doppel*) amongst a population of adults with ADHD.

## ADHD prevalence, anxiety

ADHD is the most diagnosed psychiatric disorder in children [2]. Follow up research indicates that ADHD persists into adulthood in around two thirds of cases [3]. A meta-analysis [4] suggests that the pooled prevalence of ADHD in adults is around 2.5%. From responses in 20 countries, the WHO (World Health Organization) Mental Health survey estimates a prevalence of 2.8%, with a range of 3.6% in high income countries to 1.4% in lower income countries [5]. Despite this, the European Network of Adult ADHD suggest that it is both underdiagnosed and undertreated in many European countries [3]. Research notes that adult ADHD is distinct from childhood ADHD in that an adult can modify their daily routine to better suit their needs [6]. As adults with ADHD are more likely to seek treatment for themselves, these issues are more likely to be those that cause disruption to their own quality of life, such as frustration with disorganisation and difficulties being productive. It is reported that adults with ADHD score lower on self- reported quality of life than control participants [7].

Adult ADHD is highly comorbid with anxiety and is significantly associated with role impairments when comorbidities are controlled for [5]. Difficulties associated with ADHD can take various forms. Adults with ADHD performed more poorly on tests of verbal learning and memory and sustained attention [8]. This impairment could be predicted by situational anxiety. Thus, impairments in performance shown by adults with ADHD may be influenced by the experience of anxiety. Indeed, almost 80% of adults with ADHD present with at least one lifetime psychiatric comorbidity. The odds ratio for developing an anxiety disorder in adults with ADHD is between 1.5–5.5 [9]. College students with ADHD were more likely to report a lifetime history of anxiety disorders than matched peers without a diagnosis of ADHD [10]. Moreover, they were more likely to report experiencing anxiety symptoms. These anxiety related symptoms may, in turn, lead to performance impairments in everyday tasks for those with an ADHD diagnosis.

## Management approaches and the role of wearable technology

Both American and UK (United Kingdom) treatment guidelines recommend that treatment for adults with ADHD use a combination of medication and complimentary psychoeducation and support [11]. The focus of the latter is to develop the necessary skills for structuring one's daily living. The aim of psychological intervention for adults with ADHD is to improve the core sense of self, change habitual modes of behaviour and to teach techniques that will allow the individual to control the symptoms of ADHD [12].

Given the importance of psychoeducation and symptom-management in adults with ADHD, it raises the question as to whether digital health technologies can be of benefit, in reducing the associated symptoms of anxiety reported by adults with ADHD. Digital health technology is a rapidly emerging field allowing the everyday population to manage their own physical and mental health and lifestyle using smartphone apps and wearable devices. One in six consumers in the United States report using wearable technology [13]. A prototype wearable device was tested to aid those with ADHD and attention deficiencies in maintaining sustained focus through the Pomodoro technique [14]. This is a time management technique that breaks work into short intervals with regular short breaks. Initial user testing of the prototype

demonstrated that 80% of users reported a reduction in stress and anxiety levels after engaging in each meditation session. Research conducted with children and adolescents with ADHD demonstrates the potential for wearable devices to increase physical exercise whilst reducing ADHD symptom severity [15] and improve symptom severity through tracking and providing feedback about ADHD symptoms and user's movement [16]. These findings suggest that digital health products may be beneficial in reducing symptom severity and improving daily functioning in those with ADHD. Such technologies, including apps, often make use of visual and auditory domains using prompts and feedback. However, there is an emerging body of research looking into technologies using haptic modalities.

One such study describes the development of a prototype anxiety aid using the tactile domain [17]. It presents a huggable cushion that simulates breathing. The cushion was compared to a guided meditation and a no intervention control in a social anxiety inducing situation in the form of a verbal mathematics test in front of peers. The results indicated that holding the cushion during the anticipatory phase of the study resulted in significantly lower state anxiety scores than participants in the control group. The scores were not significantly different from those in the guided meditation condition, which suggests that haptic stimulation can be as beneficial as guided meditation in reducing anticipatory anxiety. Similarly, research found that a vibrating breathing pacer reduced state anxiety when taking part in a cognitive task compared to controls [18]. There is also evidence for the efficacy of haptic feedback resembling a slower heart rate in reducing anxiety [19,20]. Thus, there is emerging evidence that digital technologies using haptic stimulation may be effective in reducing anxiety.

One digital health product that utilises haptic stimulation is *Doppel*, a wrist worn wearable. It provides the wearer with heartbeat like vibrations to the wrist. Slow vibrations are calming whilst faster vibrations increase focus. *Doppel* has been previously shown to reduce anxiety amongst a general population; In a study [21], wearers of the *Doppel* had reduced physiological and self-reported anxiety in anticipation of public speech as compared to users who wore a placebo, inactive *Doppel*. Moreover, a white paper tested the effects of a *Doppel* prototype on alertness [22]. Participants completed a psychomotor task measuring sustained attention whilst wearing the *Doppel* device on their wrist. For half of the trials, the *Doppel* delivered vibrations at a frequency of 100–120 bpm, for the second half, no vibrations were received. Participants committed significantly fewer lapses in attention when the *Doppel* was active as compared to when it was not. Thus, it suggests that the *Doppel* significantly increased participants' alertness. Doppel uses principles of biofeedback in which external stimulus is used to enable a person to develop greater awareness of their own bodily functions (for instance heart rate). A systematic review reports positive results of biofeedback interventions for anxiety [23]. Moreover, research reports the efficacy of a wearable device for Heart rate variability (HRV) monitoring combined with a remote stress management coach for anxiety symptom reduction [24]. As such there is an emerging body of evidence supporting the use of biofeedback interventions for self-management of anxiety.

Research conducted so far indicates that *Doppel* is effective in reducing anxiety and increasing alertness depending on the vibrations administered. However, these findings were based upon participants in the general population in laboratory conditions. As yet, Doppel has not been tested in conditions of everyday use. Moreover, given that anxiety and lack of focus are two concerns associated with ADHD, it begs the question, can *Doppel* help in reducing anxiety and increasing focus in adults with ADHD specifically? Anecdotal reports from users with ADHD suggest it may have a beneficial effect among this user population. As such, the present study aims to test whether the *Doppel* device and app are effective in reducing anxiety and increasing focus in young adults with ADHD when used over an eight week trial duration. The present study consisted of a double blind randomised controlled trial in which participants

were randomly assigned to use either an active *Doppel* or a comparator device for a period of eight weeks. Measures of anxiety and focus were taken at baseline and four- and eight-week time points. Based upon previous findings [21] it was predicted that there would be a greater change in anxiety and focus on young adults who have used an active *Doppel* for eight weeks as compared to those who have used a comparator device. Hypotheses, methods, and analysis plan were pre-registered on the Open Science Framework (OSF, https://osf.io/k6h7r).

## Method

### Participants

Participants were young adults between the ages of 18–25 who self-reported a formal diagnosis of ADHD. Due to the nature of the study, an additional inclusion criterion required participants to have access to a smartphone. Exclusion criteria were sensitivity to stainless steel and currently experiencing mental health difficulties for which they were taking medication (aside from ADHD). An apriori power analysis indicated that to detect a medium effect size (f = .65) between two conditions at two time points with a power of .95 and one covariate, a sample of 33 completed cases are required at eight weeks. 49 participants completed the baseline questionnaire, including 13 males, 35 females and one participant who identified as non-binary (mean age = 21.45, SD = 2.16, range = 7). Of the sample at baseline, 10.2% reported their highest level of educational qualification as GCSE or equivalent, 51% as A-level or equivalent, 24.5% undergraduate and 14.3% as postgraduate. At four weeks, 37 participants completed the four week follow up, while at 8 weeks 32 participants (14 in the experimental condition and 18 in the control condition). The study had an attrition rate of 34.69%. The study received ethical approval from the ethics panel at London South Bank University.

### Materials

**Self-report measures.** The study took measures of ADHD, sustained focus, and anxiety via a survey link at baseline, four- week and eight-week timepoints

*ADHD.* To measure ADHD the adult ADHD self-report screening scale [25] was used. This scale consists of six questions that are predictive of ADHD symptoms. Participants are required to report the frequency of these symptoms occurring on a five-point scale from zero–four, with zero being never and four very often. The maximum score of 24 indicates a greater frequency of symptoms. Cronbach's alpha = .54.

*Sustained focus.* The Everyday Attention to Life Scale [26] was administered to measure sustained focus. This scale measures participants' attentional capacities for everyday situations. It measures sustained, focused, selective, and divided attention in addition to motivation. For each situation presented, the participant is asked to report the number of minutes they could concentrate on the task (from 0–120), the percentage of their focus on the task when there are different distractions around them and their motivation to perform the task well. For each scenario, the minutes of concentration are divided by 1.2 to calculate a percentage after which the responses are averaged to get a situation score. The Cronbach's alpha for this scale was $\alpha = .57$.

*Anxiety.* The State- Trait Anxiety Inventory was used to measure self-report anxiety [27]. This questionnaire distinguishes between the more stable personality trait of anxiety proneness and the situational response of anxiety. It consists of twenty statements for each component asking participants how they feel in the present moment and in general. Both components are scored from one to four, providing two scores from 20–80, one of state anxiety and one of trait anxiety. Higher scores indicate higher self-reported anxiety. The state portion of the STAI had a Cronbach's alpha of .91while the trait portion had a Cronbach's alpha of .90.

*Acceptability and adherence*. At four- and eight-week time points, participants were also asked to report the number of days in the past week they had used their Doppel, average hours of use per day and if there was any reason they had not used it. The latter was recorded using a free text option to enable participants to feedback their experience.

## Wearable Device and accompanying smartphone app

*Doppel* is a wearable device worn on the inside of the user's wrist, positioned where the user would expect to be able to feel their pulse. The base is made of stainless steel with a plastic top, measuring 37.5mm in diameter and 8.5mm in height with a medical grade silicone strap. It uses touch controls but has no digital display.

The *Doppel* is additionally controlled via a smartphone app, where users can change the frequency of vibrations. Initially, the user can use the app to measure their own resting heart rate. Thus, the user can choose a vibration pattern that is either faster or slower than their own heart rate.

*Study apps*. Two purpose-built apps were used for the study, one for those in the experimental condition and one for those in the comparator condition. Participants in the experimental condition experienced the standard functionality of the device and app, in which they could choose the frequency of vibrations based upon their own resting heart rate. Whilst those in the comparator condition used an identical device that was programmed to deliver three vibration settings that were fixed at 0.2, 0.5 or 1 BPM.

## Procedure

**Baseline measures.** Participants were recruited via social media and support groups for adults with ADHD. Those interested in taking part were directed to an online data collection platform (Qualtrics) to provide written consent to take part and complete the baseline study measures. Participants were asked to confirm their eligibility and create a unique identifying code before completing the STAI, ELAS and ASRS scales. Additionally, participants gave a postal address for the purposes of providing the device, and an email for the purpose of providing the accompanying app to the device. Once participants had completed the baseline measures, they were randomly assigned to one of two device conditions (standard vs. comparator). The study used a randomised double-blind design, in which participants would be sent one of two app links to use with their device, with neither the participant nor the researcher aware of which condition was associated with which app link while the trial was underway. Once participants had been posted their device, they were sent an email containing (i) their link to download the app (ii) a product user guide containing the key information required to set up and use the *Doppel* (iii) information on how to contact *Doppel* in the event of any technical difficulties.

**Doppel device.** When the device arrived, participants were instructed to take some time to review the product user guide to ensure they were happy with how to use the device. They were asked to wear the device during waking hours, removing it to shower or bath. Participants were required to download the app via the link provided over email by the research team, and connect the device to the app. This enabled participants to control the device using the app or the controls on the device itself.

In the active condition, the vibration rhythm most commonly chosen was the 'relax rhythm'. This is set to be 75% of the user's resting heart rate (34.7% of choices). The 'focus rhythm', which is set to 95% of the user's resting heart rate was chosen in 23.7% of cases, whilst the 'calm rhythm' which is set to 80% of the user's resting heart rate was chosen in 24.7% of cases. In the comparator condition, the vibration pattern most commonly chosen was the

'focus rhythm' (28.6% of cases), which was set to 0.2 BPM, the 'calm rhythm' was selected in 25.4% of cases and was set to 0.5 BPM whilst 'relax' was chosen 17.9% of cases and was set to 1BPM.

**Self-report survey follow up.** At four- and eight-week time points, participants were contacted by the research team to complete follow-up measures. As with the baseline measures, participants were required to complete STAI, ELAS and ASRS scales. After completing the eight-week follow up, participants were sent a debrief form via email, and were informed that they had the option of returning the device to receive their payment for participation or keeping the device in lieu of payment. Participants that chose to receive payment had their device collected via courier, after which they were paid for their participation via PayPal.

**Design and analysis plan.** The study was a double-blind randomised controlled trial conducted on an intention to treat basis. It compared participants given a standard Doppel device with a placebo device. A standard device provided vibrations that can be increased or decreased in frequency by the user to create a calm or focused response based upon the user's resting heart rate. A placebo device provided vibrations in a fixed pattern. The primary outcome measures consisted of anxiety and sustained focus whilst secondary outcomes were self-reported satisfaction with the device and usage data. Participants were followed for eight weeks, with measures being taken at baseline (prior to device use) four- and eight-week time points.

## Results

### Current ADHD management

Participants were asked to report their current method of managing their ADHD. 59.2% of participants reported using medicinal methods, 24.5% reported non-medicinal methods and 14.3% reported using both. The non-medicinal methods of condition management included exercise, lifestyle support such as using alarms, calendars, and reminders in addition to more formal means of support such as psychotherapy and counselling. Medicinal methods of condition management included various brands of Methylphenidate, Lisdexamfetamine and Atomoxetine.

### Self-report device usage

**Four weeks.** At four weeks, participants reported using their Doppel on average 5.31 days in the past seven (S.D = 1.67). In terms of hours of use, participants reported on average 6.98 hours of use per day (S.D = 4.12).

**Eight weeks.** Participants self-report usage at eight weeks averaged 4.94 days (SD = 1.90) over the past seven days. In terms of hours of use, participants averaged 6.78 hours of use per day (SD = 3.37).

**Anxiety and focus scores at four and eight weeks.** A between subjects ANCOVA indicated that there was no significant difference in state anxiety scores between participants in the active or comparator Doppel conditions at four week follow up when controlling for baseline ADHD symptom severity $F(1,34) = 1.05$, $p =. 313$, $\eta^2 =. 03$. Additionally, there was no significant difference in focus scores between participants in the active or comparator Doppel conditions at four weeks when controlling for baseline ADHD symptom severity $F(1, 33) = .43$, $p = .517$, $\eta^2 = .013$.

At eight weeks, a between subjects ANCOVA, controlling for baseline ADHD symptom severity indicated no significant difference in state anxiety scores between the active and comparator Doppel conditions $F(1, 30) = 0.39$, $p = .525$, $\eta^2 = .013$. there was also no significant

**Table 1. Means and standard deviations (in parentheses) of anxiety and focus scores at baseline, four weeks and eight weeks.**

| | Focus score | | | State anxiety score | | |
|---|---|---|---|---|---|---|
| | Baseline | 4 weeks | 8 weeks | Baseline | 4 weeks | 8 weeks |
| **Active Doppel** | 37.92 (8.36) | 43.43 (9.42) | 45.63 (10.07) | 55.29 (10.44) | 47.00 (8.94) | 42.27 (11.49) |
| **Comparator Doppel** | 38.78 (9.04) | 44.37 (10.28) | 45.97 (15.93) | 57.12 (10.37) | 44.23 (10.27) | 43.72 (12.51) |
| **Total** | 38.36 (8.63) | 44.00 (9.83) | 45.82 (13.48) | 56.23 (10.34) | 45.35 (8.47) | 43.06 (11.90) |

difference in focus scores between Doppel conditions at eight weeks, when controlling for baseline ADHD symptoms severity $F(1, 29) = 0.009$, $p = .923$, $\eta^2 < .001$.

Thus, wearing an active Doppel for eight weeks did not decrease state anxiety or increase focus as compared to a comparator Doppel. Means and standard deviations are displayed in Table 1.

**Yoked analysis.** As per our pre-registered analysis plan, we also conducted a yoked analysis. Participants in each device condition were ranked by baseline ADHD severity and yoked across device condition by rank. This created pairs of participants (one from each condition) who were matched on baseline ADHD severity scores. Subsequently a within subjects' analysis on yoked pairs was conducted.

There was no significant difference within groups on state anxiety scores at 8 weeks when covarying baseline anxiety scores $F(1, 8) = 0.14$, $p = .72$, $\eta^2 = .02$.

There was also no significant difference within groups on focus scores at 8 weeks when covarying baseline focus $F(1, 8) = 0.08$, $p = .78$, $\eta^2 = .01$.

**Change in anxiety and focus over time.** A repeated measures ANCOVA was conducted to examine the effects of Doppel condition on state anxiety scores over the duration of the study (baseline, four weeks, eight weeks) with ADHD scores at baseline as a covariate. There was a significant effect of time on anxiety scores $F(2, 56) = 3.21$, $p = .048$, $\eta^2 = .10$. Anxiety was significantly higher at baseline (EMM = 56.43, SE = 1.95) than at four weeks (EMM = 44.91, SE = 1.62) and eight weeks (EMM = 43.40, SE = 2.26) ($Ps < .001$). There was no significant interaction between Doppel condition and anxiety change over time $F(2, 56) = 0.84$, $p = .437$, $\eta^2 = .03$. The covariate, baseline ADHD scores, was significantly related to participants' anxiety $F(2, 56) = 5.03$, $p = .004$, $\eta^2 = .177$

A repeated measures ANOVA indicated that participants focus scores significantly increased over time (The assumption of sphericity was violated, as such the Greenhouse Geisser correction was reported). $X^2(2) = 8.88$, $p = .012$. $F(1.57, 45.60) = 8.09$, $p = .002$, $\eta^2 = .22$. Participants focus scores were significantly lower at baseline (EMM = 39.12, SE = 1.56) than four weeks (EMM = 44.09, SE = 1.84) ($p = .003$) and eight weeks (EMM = 45.64, SE = 2.53) ($p = .012$). There was no significant interaction between Doppel condition and focus scores over time $F(1.57, 45.60) = .03$, $p = .94$, $\eta^2 = .002$. Thus, anxiety and focus improved over the eight-week trial duration, however this did not differ between the active and comparator Doppel conditions.

**Exploratory analysis.** We conducted an exploratory analysis to identify any effects of self-report usage on state anxiety and sustained focus across both device conditions. A repeated measures ANOCVA with self-report device usage at 8 weeks found controlling for self-reported device usage, there was a significant effect of time $F(2, 50) = 3.32$, $p = .044$, $\eta^2 = .12$. Participants were significantly less anxious at time three than time one, and at time two than time one ($ps < .01$). There was no significant relationship between self-reported usage and anxiety ($p = .774$).

When controlling for self-reported device usage, the main effect of time on focus scores was not significant, $F(2, 50) = 1.82$, $p = .18$, $\eta^2 = .07$. However, participants were significantly more

**Table 2. Coefficients of predictors in moderation models for anxiety and focus.**

|  | B | SE B | t | sig |
|---|---|---|---|---|
| **Anxiety** |  |  |  |  |
| **Usage** | .72 | 2.17 | .33 | .74 |
| **Doppel condition** | 3.71 | 10.14 | 3.67 | .72 |
| **Baseline anxiety** | .40 | .22 | 1.85 | .08 |
| **Usage x Doppel** | -.29 | 1.27 | -.23 | .82 |
| **Focus** |  |  |  |  |
| **Usage** | .45 | 2.48 | .18 | .86 |
| **Doppel condition** | 8.85 | 11.50 | .77 | .45 |
| **Baseline focus** | .31 | .28 | 1.11 | .28 |
| **Usage x Doppel** | -.95 | 1.45 | -.65 | .52 |

focused at time two than time one, and at time three than time one ($p$s < .05). There was no significant relationship between device usage and focus scores ($p$ = .89). Thus, when controlling for device usage, the pattern of results stays the same.

Finally, an exploratory analysis was conducted to examine whether the relationship between device usage and anxiety was moderated by device condition. This model was computed using a regression-based approach [28]. The moderation model chosen examined whether device condition (active or comparator Doppel) moderated the relationship between device usage and anxiety at eight weeks, when covarying baseline anxiety scores.

The model predicting anxiety was not significant. $F(4, 23)$ = 0.99, p = .43, $R^2$ = .15. Usage and Doppel condition predicted 15% of the variance in anxiety and were not significant independent predictors of anxiety. The interaction between device usage and condition was also not significant ($p$ = .82). Therefore, anxiety scores at eight weeks were not predicted by Doppel usage, and this relationship was not moderated by whether participants received an active or comparator Doppel.

The same model was also computed for focus scores. The model predicting focus was not significant $F(4, 23)$ = 1.05, $p$ = .40, $R^2$ = .15. Usage and Doppel condition predicted 15% of the variance in focus scores and were not significant independent predictors. The interaction between usage and Doppel condition was also not significant ($p$ = .52). Thus, focus scores at eight weeks were not predicted by Doppel usage, and this relationship was not moderated by whether participants received an active or comparator Doppel. Table 2 presents the coefficients for both models.

**Acceptability, Discontinuation and trouble shooting of intervention.** Participants were given free text options to report whether i) there were any reasons they had not worn their Doppel and ii) if they had experienced any problems as a result of wearing the Doppel.

Participants' responses were reviewed for common themes, which are outlined below. 9 participants reported experiencing problems as a result of wearing their Doppel at 4 and 8 weeks.

**Hardware and tech issues.** Participants reported experiencing difficulties specifically with the strap. It was reported that the strap would break or feel fragile or the Doppel would come loose. Some participants reported experiencing technical issues with the Doppel and accompanying app. A common issue that emerged for participants was the tendency for the Bluetooth to disconnect, which meant they had to reconnect the device. One participant mentioned that when the device disconnects and stops, this causes them to become distracted. Participants also reported finding that the battery life was not as long as expected.

**Activity specific considerations.** Some participants reported not using their Doppel for specific activities. This included lab work which requires maintaining bare arms and engaging in sports to avoid breaking it.

**Mood.** Participants also discontinued use for a range of mood factors. One of these included reporting that the device vibrations caused them irritation. Some participants provided reasons more specific to the experience of living with ADHD. These included finding that it added to sensory overload, that it was distracting when one is 'hyperfocused' and that it leads to anxiety for one person when they are medicated as the vibrations mimic their racing heart.

**Forgetting.** Participants reported not using their Doppel due to i) forgetting to charge their device and ii) forgetting to put on the Doppel.

## Discussion

The present study aimed to identify whether a wrist worn wearable (Doppel) could reduce ADHD symptoms (anxiety and low levels of focus) by delivering vibrations based upon user's heart rate, over an eight-week trial period. This level of symptom change was also compared to a control condition in which vibrations were emitted according to a wearer selected schedule that was not linked to users' heart rate. Overall, the findings demonstrated a decrease in both symptoms over time. However, neither schedule of vibration patterns demonstrated superiority in the size of these decreases.

The present findings add to an emerging body of literature examining the potential for wearable devices to be used for the self-management of health outcomes [13]. In particular, they complement previous research into Doppel in other domains [21], which demonstrated a reduction in skin conductance and anxiety in anticipation of public speaking amongst participants wearing a Doppel device and receiving heart rate calibrated vibrations as compared to a control group who wore a Doppel device that was turned off. However, the current set of findings (equal symptom improvement in both vibration schedule conditions) suggest that the beneficial element of the device may not be the heart rate matching component (in this population, for these symptoms). This could imply that receiving vibrations at a fixed schedule may offer ADHD symptom relief regardless of whether these vibrations are linked to heart rate. In support of this possibility, research [14] found that users with ADHD reported finding inhale/exhale vibration cues administered via a smartwatch app helpful during meditation sessions. Moreover, the benefits of haptic feedback for those with ADHD has further been proposed in research that worked with adolescents with ADHD to develop a tool to aid in their perception of the passage of time [29]. Participants requested a haptic component in the form of continuous vibration, to act as an external cue to the passing of time. Further research is needed to elucidate the underlying processes in Doppel's positive effects on anxiety and focus amongst with ADHD. However, taken together with research in other domains, the findings point towards a benefit of haptic stimulation generally, and the Doppel device specifically, for those with this condition.

Treatment for ADHD in adults often includes the use of self-management techniques [12]. A key consideration of Doppel's utility to support those with ADHD is its tolerability amongst users, in addition to any beneficial effects. The high self-reported adherence amongst participants in the trial (Av. 7 hours per day) suggests that Doppel is an acceptable intervention for adults with ADHD. This, combined with a single and relatively low intervention cost (current direct to consumer retail price of the product is £175) provides a potentially scalable intervention in both public and private health settings.

## Key study limitations

There are some limitations to the study that should be considered. First, the study deployed the use of an active comparator condition. Participants in the comparator condition were still able to choose the pattern of vibrations they received; however, these were not based upon participants' heart rate. As such, the findings tell us that receiving scheduled vibrations may improve focus and reduce anxiety, but do not demonstrate the superiority of having a schedule of vibrations that are based on users' heart rates. Future research could include fixed intermittent vibrations as a more passive comparator condition with which to compare the standard Doppel to.

Whilst self-report usage was generally high (av. 7 hours per day) it was not possible to measure actual device usage from the data collected by the Doppel app. As such, the usage data suffers from the issues inherent in self-reporting and may not be an accurate reflection of participant's actual usage of their Doppel. Moreover, whilst a significant proportion of participants were using some form of treatment for their ADHD during the trial, we feel this is not inconsistent with how Doppel could be used in the real world. Doppel is designed to promote calmness and increase focus rather than as a treatment for ADHD. As such it would likely be used as an adjunct to treatment for additional symptom management

The environmental context in which the research took place should also be considered. Data were collected during the global COVID-19 pandemic. During the data collection process, the UK went through various stages of lockdown restrictions. This made recruitment more difficult. The trial also experienced quite significant drop-out which may have been influenced by the follow-ups taking place via online survey sent over email. It is also possible that the reduction in anxiety may at least in part, be influenced by the changing lockdown conditions over the trial. However, the study did also find an increase in focus over the course of the trial which is unlikely to be attributable to this cause. Moreover, the length of the recruitment and data collection period included periods of both diminishing and increasing intensity of the pandemic and associated restrictions, press coverage, etc, in a non-linear fashion. In contrast, the impact of the Doppel device for both symptom sets decreased across both time-points consistently.

The present study also aimed to measure the acceptability of the intervention by measuring average hours of usage per day, number of days of usage per week, reasons for discontinuing use and issues encountered. This enabled us to understand behaviours associated with acceptability (usage) and attitudes (discontinuing use and issues encountered). However, we could have additionally included questions on participants' perceptions of the device, affective response and more general reviews.

## Future directions

Future work should consider the potential for the integration of these vibration schedules within existing smart tech devices and/or distinguishing for whom heart rate matched schedules provide additional benefits. Participants' most frequent issue centred around hardware problems. As such, addressing the specific issues and/or incorporating the technology into an existing piece of hardware may resolve some of these barriers to use. Moreover, it could be investigated whether the effects can be drawn from smart watches or if the effects exist even if delivered through a smart phone for example.

Future research is also needed to pinpoint the nature of the benefits the Doppel provides. A comparison of active, active comparator and no device conditions for instance would provide evidence as to whether receiving heart rate timed vibrations offers superior benefits compared to a fixed vibration pattern or a no device comparator.

## Conclusion

The present study examined the effects of Doppel, a wearable device, on levels of anxiety and focus on young adults with ADHD over the course of an 8-week trial period. The findings do not demonstrate a superiority of symptom improvement between the standard Doppel vibration schedule over the comparator schedule. However, they do suggest that receiving vibrations to the wrist confers benefits to the user, regardless of whether these vibrations are linked to heart rate. The findings provide promising initial evidence that a wrist-worn wearable is an acceptable and scalable intervention for young adults with ADHD, with the potential to improve the symptoms of anxiety and poor focus often reported by those with ADHD [9,6].

## Author Contributions

**Conceptualization:** Daniel Frings, Eddie Chaplin.

**Formal analysis:** Georgina Bartlett, Daniel Frings.

**Methodology:** Georgina Bartlett, Daniel Frings, Eddie Chaplin.

**Supervision:** Daniel Frings, Eddie Chaplin.

**Writing – original draft:** Georgina Bartlett.

**Writing – review & editing:** Georgina Bartlett, Daniel Frings, Eddie Chaplin.

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
