## [Decision Letter · Decision Letter 0]

12 Feb 2024

PDIG-D-23-00319

Evaluating Doppel's impact on Anxiety and Focus amongst adults with ADHD

PLOS Digital Health

Dear Dr. Bartlett,

Thank you for submitting your manuscript to PLOS Digital Health. After careful consideration, we feel that it has merit but does not fully meet PLOS Digital Health's publication criteria as it currently stands. Therefore, we invite you to submit a revised version of the manuscript that addresses the points raised during the review process.

Please submit your revised manuscript within 60 days Apr 12 2024 11:59PM. If you will need more time than this to complete your revisions, please reply to this message or contact the journal office at digitalhealth@plos.org. Please include the following items when submitting your revised manuscript:

We look forward to receiving your revised manuscript.

Kind regards,

Mengling Feng

Academic Editor

PLOS Digital Health

Journal Requirements:

1. Please send a completed 'Competing Interests' statement, including any COIs declared by your co-authors. If you have no competing interests to declare, please state "The authors have declared that no competing interests exist". Otherwise please declare all competing interests beginning with the statement "I have read the journal's policy and the authors of this manuscript have the following competing interests:"

If you did not receive any funding for this study, please simply state: “The authors received no specific funding for this work.

Additional Editor Comments (if provided):

Reviewers' comments:

Reviewer's Responses to Questions

**Comments to the Author**

1. Does this manuscript meet PLOS Digital Health’s publication criteria? Is the manuscript technically sound, and do the data support the conclusions? The manuscript must describe methodologically and ethically rigorous research with conclusions that are appropriately drawn based on the data presented.

Reviewer #1: Yes

Reviewer #2: Partly

Reviewer #3: Partly

2. Has the statistical analysis been performed appropriately and rigorously?

Reviewer #1: Yes

Reviewer #2: N/A

Reviewer #3: Yes

3. Have the authors made all data underlying the findings in their manuscript fully available (please refer to the Data Availability Statement at the start of the manuscript PDF file)?

Reviewer #1: No

Reviewer #2: Yes

Reviewer #3: Yes

4. Is the manuscript presented in an intelligible fashion and written in standard English?

Reviewer #1: Yes

Reviewer #2: Yes

Reviewer #3: Yes

5. Review Comments to the Author

Reviewer #1: Overall, this is a clear, concise, and well-written manuscript. The introduction is relevant and sufficient background information provided. The methods, results, and discussions are consistent and appropriate. The results are also described with greater clarity. My only concern is how “Acceptability” is being conceptualised and measured in this study. 

1. The authors claimed to have measured acceptability of the intervention by asking open-ended questions on whether 1) participants had any reasons for not wearing doppel and 2) if they had experienced any problems as a result of wearing the Doppel. I don’t think these accurately measures acceptability of an interventions. I see these questions as more of assessing the participants’ view of the intervention or self-reported barriers in using the intervention and the authors should be clear if that is what they mean. 

2. Also, the acceptability aspect should be included in the methods section. Only the STAI, ELAS and ASRS scales are included in the self-report survey follow-up and not the interview questions. 

Here are some minor comments

1. Spell out the acronym for ADHD

2. The literature review in the background shows that doppel has been shown to reduce anxiety in previous studies. If the intervention has been shown to reduce anxiety is other studies, then there is high likelihood that the result might be replicated in your study – proving your hypothesis right. To get a compelling argument as to why you need another study on the same intervention, I think the authors should provide a justification as to why this study is needed in the ADHD population.

Reviewer #2: The sample size of 49 participants appears too small for a reliable evaluation of the proposed outcomes. Furthermore, the experiment concluded with a total of 32 participants, without clearly specifying the number of participants in each group (Doppel or comparator). Based on the author's statement that "59.2% of participants reported using medicinal methods, 24.5% reported non-medicinal methods, and 14.3% reported using both," it appears that 73.5% of participants were under some form of medication or intervention. This makes it challenging to accurately assess the effectiveness of the Doppel intervention.

Reviewer #3: This work presents the results of an 8-week trial of using Doppel, a wearable technology providing vibrations based on the wearer's heart rate.

While the use of digital technology for managing anxiety and increasing focus is motivated well, the work does not adequately review technologies that use bio-feedback such as Doppel does. Doppel has been evaluated previously with a relatively small general population sample in a simulated task with an active and inactive device; this work investigates Doppel with an active device and a comparator device in young adults with ADHD over the course of 8 weeks.

The protocol was pre-registered with and is decribed relatively well. There are several points to note about the participants: there is a low sample size (not quite meeting the minimum sample size necessary at week 8) and a surprising amount of female participants. Ethical approval was given but is unnecessarily blinded.

The instruments used to measure anxiety and focus seem appropriate but it is unclear how they were administered and when. The device is decribed well but it was difficult to understand how the active and comparator device differed and why these different setting were implemented in the way they were. At first I thought that the comparator device was providing randomised vibrations but instead it turns out that the vibrations are at static BPM. It seems users could choose which rhythm to use, and I wonder if this had any effect on the measures or how that was accounted for in the analysis. For use, self-reported measures were employed but could this be more objectively measured using app logs?

The analysis is fine but I have severe doubts about the robustness of the measures. Similarly, there does not seem to be any difference between active and comparative device in anxienty or focus - wearing either one leads to a decrease in anxiety and an increase in focus over 8 weeks. Hence, I would say that both conditions are a "placebo" in a way whereas this work suggests that there is "some promising initial evidence as to the potential for direct-to-consumer digital health products to be useful in symptom management amongst young adults with ADHD". I strongly suggest to qualify this, especially given the retail price of Doppel.

6. PLOS authors have the option to publish the peer review history of their article (what does this mean?). If published, this will include your full peer review and any attached files.

**Do you want your identity to be public for this peer review?** For information about this choice, including consent withdrawal, please see our Privacy Policy.

Reviewer #1: No

Reviewer #2: No

Reviewer #3: No

---

## [Decision Letter · Decision Letter 1]

18 Jun 2024

Evaluating Doppel's impact on Anxiety and Focus amongst adults with ADHD

PDIG-D-23-00319R1

Dear Dr Bartlett,

We are pleased to inform you that your manuscript 'Evaluating Doppel's impact on Anxiety and Focus amongst adults with ADHD' has been provisionally accepted for publication in PLOS Digital Health.

Best regards,

Daniel B. Forger

Section Editor

PLOS Digital Health

Reviewer Comments (if any, and for reference):

Reviewer's Responses to Questions

**Comments to the Author**

1. If the authors have adequately addressed your comments raised in a previous round of review and you feel that this manuscript is now acceptable for publication, you may indicate that here to bypass the “Comments to the Author” section, enter your conflict of interest statement in the “Confidential to Editor” section, and submit your "Accept" recommendation.

Reviewer #2: All comments have been addressed

Reviewer #3: All comments have been addressed

2. Does this manuscript meet PLOS Digital Health’s publication criteria? Is the manuscript technically sound, and do the data support the conclusions? The manuscript must describe methodologically and ethically rigorous research with conclusions that are appropriately drawn based on the data presented.

Reviewer #2: Yes

Reviewer #3: Yes

3. Has the statistical analysis been performed appropriately and rigorously?

Reviewer #2: N/A

Reviewer #3: Yes

4. Have the authors made all data underlying the findings in their manuscript fully available (please refer to the Data Availability Statement at the start of the manuscript PDF file)?

Reviewer #2: Yes

Reviewer #3: Yes

5. Is the manuscript presented in an intelligible fashion and written in standard English?

Reviewer #2: Yes

Reviewer #3: Yes

6. Review Comments to the Author

Reviewer #2: as long as this method or product using for direct-toconsumer digital health, it should be benefit for the ADHD patients

Reviewer #3: All my comments have been addressed. I have reviwed the paper and I believe that it is now ready for publication.

7. PLOS authors have the option to publish the peer review history of their article (what does this mean?). If published, this will include your full peer review and any attached files.

**Do you want your identity to be public for this peer review?** For information about this choice, including consent withdrawal, please see our Privacy Policy.

Reviewer #2: No

Reviewer #3: No
